# Increasing extreme melt in northeast Greenland linked to foehn winds and atmospheric rivers

Kyle S. Mattingly [1,2] ✉, Jenny V. Turton [3,4], Jonathan D. Wille [5,6], Brice Noël [7,8], Xavier Fettweis [8], Åsa K. Rennermalm [9] & Thomas L. Mote [10]

The Greenland Ice Sheet has been losing mass at an increased rate in recent decades. In northeast Greenland, increasing surface melt has accompanied speed-ups in the outlet glaciers of the Northeast Greenland Ice Stream, which contain over one meter of sea level rise potential. Here we show that the most intense northeast Greenland melt events are driven by atmospheric rivers (ARs) affecting northwest Greenland that induce foehn winds in the northeast. Near low-elevation outlet glaciers, 80–100% of extreme (> 99th percentile) melt occurs during foehn conditions and 50–75% during ARs. These events have become more frequent during the twenty-first century, with 5–10% of total northeast Greenland melt in several recent summers occurring during the ~1% of times with strong AR and foehn conditions. We conclude that the combined AR-foehn influence on northeast Greenland extreme melt will likely continue to grow as regional atmospheric moisture content increases with climate warming.

The Greenland Ice Sheet (GrIS) has lost ~3,900 billion tons of ice since 1992 and contributed 10.8 mm to global mean sea level rise, of which more than 50% is attributed to surface mass loss due to increased meltwater runoff[1]. In the last decade, the largest melt anomalies and overall mass loss acceleration have shifted poleward from southern to northern Greenland[2,3]. After the unprecedented ice sheet-wide melt episode in July 2012, a number of recent major melt events have disproportionately affected northern Greenland[4–6].

East of the northern Greenland orographic divide, solid ice flow is dominated by the fast-flowing Northeast Greenland Ice Stream (NEGIS). The NEGIS drains ~16% of the GrIS and terminates at three primary marine outlet glaciers (Nioghalvfjerdsbrae or 79°N, Zachariae Isstrøm, and Storstrømmen) that collectively contain over 1 m of potential sea level rise[7]. NEGIS outlet glaciers have exhibited increasing mass loss in recent years, due to warming air and ocean temperatures

leading to the loss of buttressing sea ice and ice shelf collapses at the floating glacier margins[8–13].

Ice flow dynamics in northeast (NE) Greenland are linked to surface hydrology through the extensive network of supraglacial streams and lakes[14] that forms in NE Greenland during the summer. Ice sheet surface melt events have been linked to summer speedups of 79°N Glacier, and glacier acceleration after supraglacial lake drainage at the end of the melt season has also been observed[7,15,16]. Extreme melt events decrease the ice sheet albedo and precondition the surface for enhanced subsequent melt[17,18]. Surface runoff may also be influenced by the development of near-surface ice slabs which may persist for years after short-lived extreme melt events and are found extensively in the NEGIS catchment[19]. These characteristics indicate that both the liquid runoff and solid ice flow components of NE Greenland's sea level rise contribution are influenced by warming-induced surface melt, and

[1]Space Science and Engineering Center, University of Wisconsin–Madison, Madison, WI, USA. [2]Institute of Earth, Ocean, and Atmospheric Sciences, Rutgers, the State University of New Jersey, Piscataway, NJ, USA. [3]Climate System Research Group, Institute of Geography, Friedrich-Alexander University, Erlangen, Germany. [4]Arctic Frontiers AS, Tromsø, Norway. [5]Institut des Géosciences de l'Environnement, CNRS/UGA/IRD/G-INP, Saint Martin d'Hères, France. [6]Institute for Atmospheric and Climate Science, ETH Zurich, Zurich, Switzerland. [7]Institute for Marine and Atmospheric Research Utrecht, Utrecht University, Utrecht, the Netherlands. [8]Department of Geography, University of Liège, Liège, Belgium. [9]Department of Geography, Rutgers, the State University of New Jersey, Piscataway, NJ, USA. [10]Department of Geography, University of Georgia, Athens, GA, USA. ✉e-mail: ksmattingly@wisc.edu

particularly by extreme melt events with compounding short- and long-term impacts.

Several studies have suggested a link between intense NE Greenland melt events and warm, dry downslope winds known as "foehn" descending from the GrIS plateau to the west[3,20–22]. Mattingly et al.[21] found evidence that atmospheric rivers (ARs) affecting northwest (NW) Greenland may lead to foehn conditions and enhanced melt in NE Greenland after the moist air mass crosses the ice divide and flows downslope, similar to warming documented on the lee side of the West Antarctic peninsula and other mountainous areas[23–26]. However, there has not been any systematic study of the role of foehn in forcing melt in northern Greenland, where the topography consists of a wide dome with a relatively gentle slope in comparison with the more narrow mountain ranges studied in other regions.

In this study, we utilize AR detection algorithms, a foehn identification procedure applied to output from two regional climate models, and a high-resolution atmospheric model simulation to demonstrate that foehn winds induced by ARs drive the majority of extreme melt events in NE Greenland. We first describe the substantial contribution of NW Greenland AR events to NE Greenland melting, then analyze the relationship between ARs and foehn-induced extreme melt events, and finally quantify the contribution of this AR-foehn-extreme melt mechanism to the observed increasing trend in NE Greenland summer melt.

## Results

### Intense northeast Greenland summer melt following western Greenland atmospheric rivers

The capability of ARs impinging on western Greenland to trigger melt in NW Greenland is exemplified by an intense AR landfall in NW Greenland on 20 July 2014 which triggered a downsloping foehn wind that produced significant melt in the NE region of the

GrIS (Fig. 1, Supplementary Movie S1). The orographic ascent of the AR moisture flux over the GrIS enhanced the latent heat release resulting in an intense downward sensible heat flux anomaly and a detectable foehn wind over the base of the ice sheet near Kronprins Christian Land (Fig. 1, Supplementary Fig. S1). After the melt event abated, MODIS satellite imagery showed an expanded region of bare ice along with numerous melt ponds near the glacial terminus (Supplementary Fig. S1b). Overall from 19–24 July 2014, -19 Gt of surface melt were produced in the NE GrIS according to the polar (p) Regional Atmospheric Climate Model version 2.3p2 (RACMO2) model.

Aggregating the cumulative melt amounts from all AR landfalls in NW Greenland from 1980 to 2020, the immediate melt impact of ARs is concentrated on the plateau and higher elevations of the GrIS. Here, Fig. 2 shows that 75–100% of summer (JJA) surface melt is produced during AR landfalls which occur with a seasonal frequency of 12–15% (13–16 days per summer according to the ref.[27] AR detection algorithm). Over the 48 h following an AR, the amount of melting associated with AR events increases at lower elevations to ~40–50% of total JJA melting (Fig. 2b, c). This eastward and downward shift of melting follows the region of maximum downward sensible heat flux anomalies, which reach a maximum in NE Greenland 24 h after the initial AR landfall, while downward longwave radiation anomalies generally remain elevated closer to the AR landfall location in western Greenland (Supplementary Fig. S2). The downward shortwave anomalies inversely mirror the longwave and show the foehn's cloud clearance effect in NE Greenland (Supplementary Fig. S2g–i), which is an important component of foehn related surface melt on the Antarctic Peninsula[28–30]. The impact of ARs on melting in NE Greenland is clearer when assessing the extreme melt events (>99th percentile): up to 75% of extreme melt events at low-lying elevations are associated with AR landfalls (Fig. 2d–f).

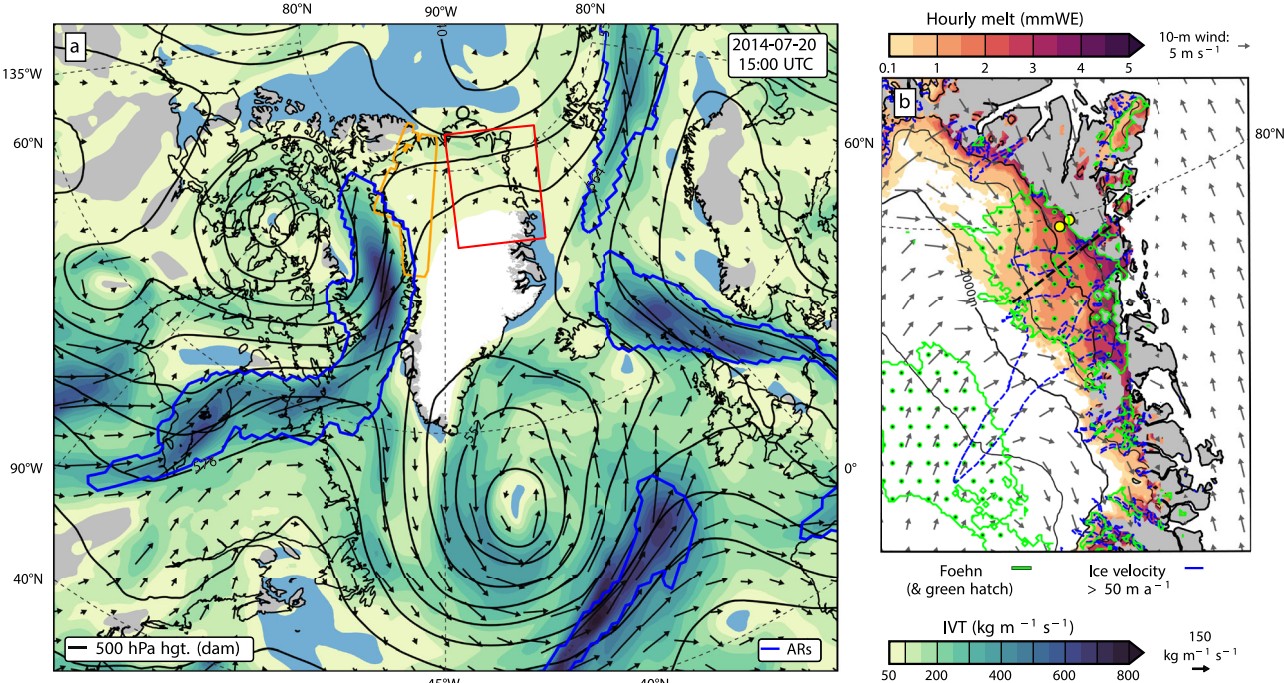

**Fig. 1 | 20 July 2014 atmospheric river (AR) and melt event. a** Map displaying outline of an AR affecting northwest (NW) Greenland (outlined in blue) from the Modern-Era Retrospective analysis for Research and Applications, Version 2 (MERRA-2) reanalysis at 20 July 2014 15:00 UTC along with integrated water vapor transport (IVT) (color shading and vector arrows) and 500 hPa geopotential height contours (black lines). Also shown are the northeast (NE) Greenland study domain (red outline), and the NW Greenland domain used for analysis of AR impacts in

subsequent figures (orange outline). **b** Hourly melt rates in the NE Greenland study region and 10-meter wind vectors simulated by the polar Regional Atmospheric Climate Model version 2.3p2 (RACMO2) along with areas of detected foehn conditions (green outline and hatching). Also shown are mean ice velocity > 50 m a⁻¹ highlighting the location of the northeast Greenland Ice Stream (NEGIS) (blue dashed contour), location of cross section analyzed in Fig. 4 (black dashed line), and the KPC_L and KPC_U weather stations used in Fig. S3 (yellow dots).

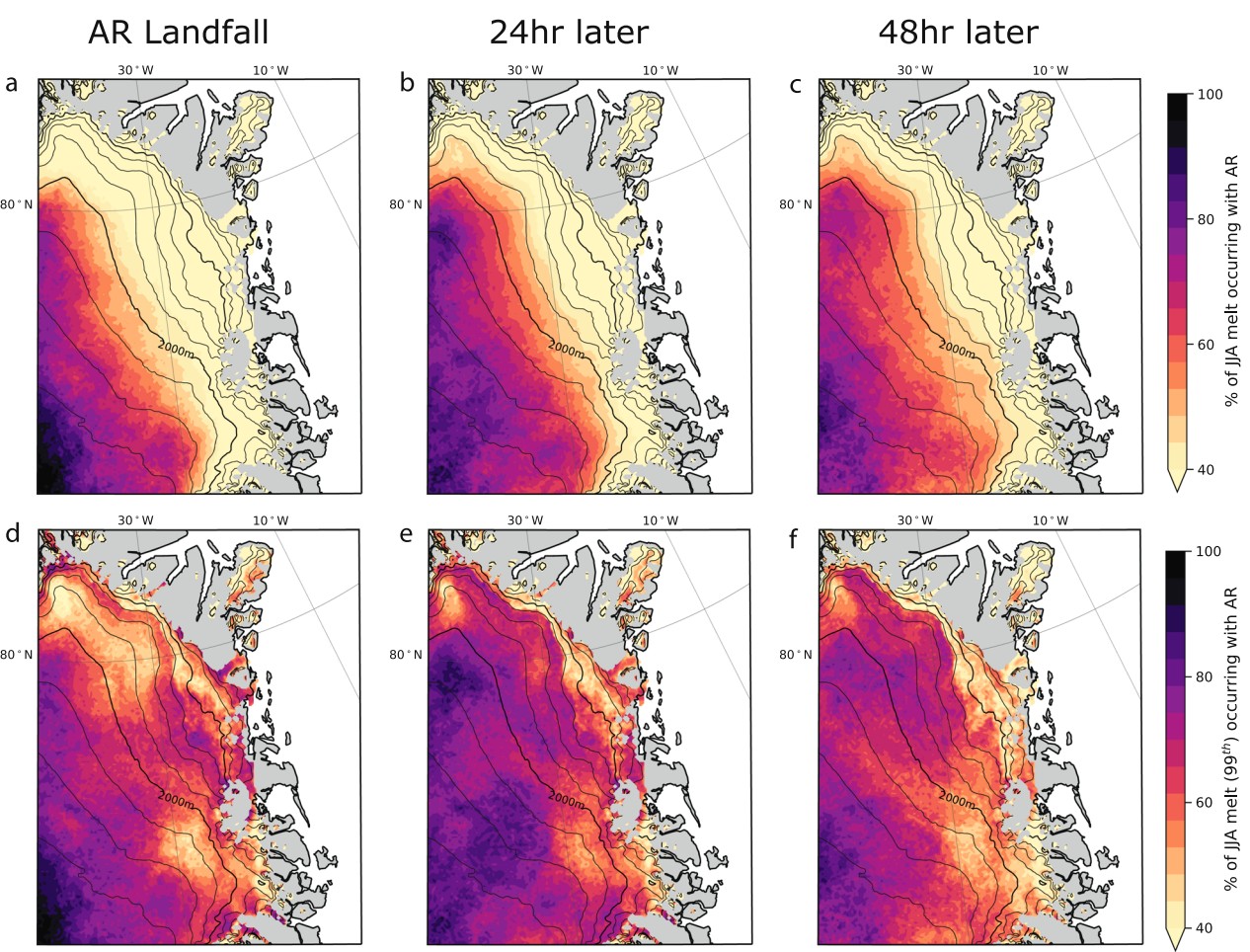

**Fig. 2 | Percentage of summer (JJA) surface melt attributable to atmospheric rivers (ARs).** **a**–**c** The percentage of polar Regional Atmospheric Climate Model version 2.3p2 (RACMO2) simulated summer (JJA) daily surface melt and **d**–**f** the percentage of surface melt at or above the 99th percentile of the 1980–2020 monthly mean climatology that occurred (**a**), (**d**) the same day, (**b**), (**e**) 24 hours later, and (**c**), (**f**) 48 h later of an AR landfall in northwest (NW) Greenland (orange outline in Fig. 1a).

The rate of melting in NE Greenland increases as a function of AR intensity, where intensity is measured by the maximum integrated water vapor transport (IVT) upon landfall (Fig. 3). ARs with a maximum IVT exceeding 500 kg m$^{-1}$ s$^{-1}$ generate a median of ~2 Gt and a maximum of nearly 8 Gt of surface melt in the 24 h after the initial AR landfall, whereas ARs with lower intensities generate lower median melt and never produce the most extreme melt amounts. This relationship ceases in months outside of summer (JJA) when melt magnitude becomes negligible. A similar phenomenon is observed over the Antarctic Peninsula where a greater IVT upon landfall leads to a greater latent heat release, as the moisture is orographically lifted over the coastal mountain ranges and creates a more intense leeward foehn wind compared to non-AR related foehn events[31].

### Downsloping foehn winds drive melt
In order to understand the physical processes by which NW Greenland ARs lead to melt in the NE, we now investigate the role of foehn winds in warming the lower troposphere. We first analyze the physical mechanisms at work during the AR event in July 2014 using atmospheric cross sections derived from a high-resolution (1 km) case study simulation of the NEGIS region using the Polar Weather Research and Forecasting (Polar WRF) model[32] (Fig. 4). During 18–19 July 2014, ARs were detected in NW Greenland, followed by intense ARs detected from 20–22 July 2014. After landfall in the NW, the AR air mass descended the eastern flank of the GrIS into NE Greenland. The adiabatic

descent of the air mass shown by the downward progression of higher potential temperature values in Fig. 4c, in combination with increased turbulence and the loss of moisture on the leeward flank of the ice sheet, produced warmer and drier conditions in the NE. This is independently confirmed by a sharp drop in relative humidity, accompanied by a rise in air temperature, observed at two automatic weather stations in the vicinity of 79°N Glacier on 19–20 July (Supplementary Fig. S3). Typical foehn conditions (dry and warm air) were continuously observed until 24 July 2014, before conditions returned to near climatological averages.

Figure 4d also shows stably stratified, persistent cold air pooling over the tongue of the 79°N Glacier and adjacent sea ice, which appeared to limit the spatial extent of the warm foehn air at very low elevations. However, we hypothesize that the amount of foehn-driven warming and melt may be underestimated over the floating glacier tongue due to the inability of the relatively coarse model to simulate the erosion and dissipation of the cold pool at lower elevations, as found by prior model studies in NW Greenland[33] and Alpine valleys[34]. Alternatively, this could be a real effect related to lee waves associated with the foehn effect drawing cool air from the east, which could be investigated with more detailed simulations in future studies.

Having demonstrated the role of foehn winds in forcing NE Greenland melt during one example of a strong summer AR event in NW Greenland, we now examine the prevalence of foehn-induced melt and its association with ARs during the full climatological record

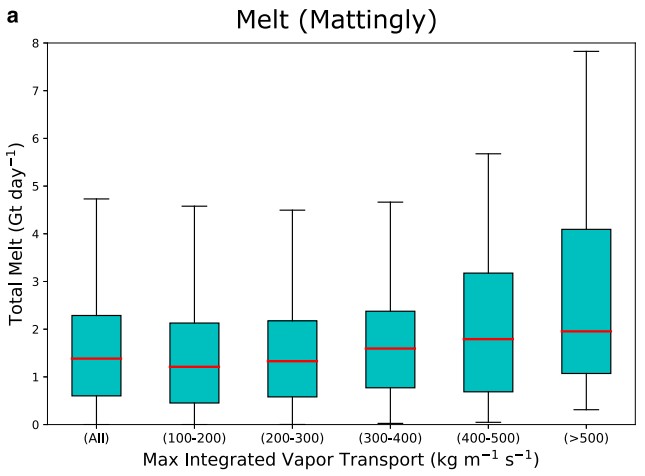

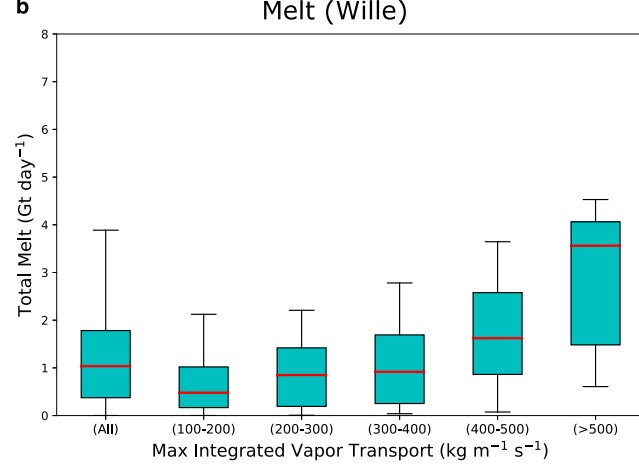

**Fig. 3 | Relationship between summer (JJA) atmospheric river (AR) intensity and magnitude of surface melt.** Box and whisker plot shows the maximum integrated water vapor transport (IVT) of northwest (NW) Greenland landfalling ARs (orange box in Fig. 1a) detected using the algorithm in (**a**) Mattingly et al.[21] and (**b**) Wille et al.[31] and the corresponding daily surface melt 24 h after the initial AR landfall summed over the northeast (NE) Greenland domain (red box in Fig. 1a). The boxes represent values from the first to the third quartile, the red line is the median, and the whiskers extend to the data maximum and minimum. Data sources are the Modern-Era Retrospective analysis for Research and Applications, Version 2 (MERRA-2) reanalysis for IVT and the polar Regional Atmospheric Climate Model version 2.3p2 (RACMO2) for surface melt.

(1980–2020). Significant surface melt in NE Greenland is primarily confined to the narrow, low-elevation (<1000 m) ablation zone along the ice sheet margins (Fig. 5a). The exception is the middle and lower reaches of the concave, bowl-like NEGIS catchment, where substantial average melt rates extend far inland and up to 1500 m elevation. This corresponds to the area where the greatest proportion of melt occurs during foehn conditions, with 35–50% of melt in the lower NEGIS basin occurring during the 25–40% of time when foehn conditions occur (Supplementary Fig. S4).

Foehn conditions play a particularly important role in driving melt during NW Greenland AR events and are responsible for most extreme melt events in the NE. During the 48 h after strong (>90th percentile IVT) NW Greenland AR landfalls (hereafter termed $AR_{90}$ events), the majority of melt (50–65%) in the middle and lower NEGIS catchment occurs during foehn (Fig. 5b). Foehn conditions also frequently generate melt along the lee slopes of outlying ice caps detached from the main ice sheet, most notably on the large Flade Isblink Ice Cap. At all elevations, there is a delay of 18–24 h between AR landfall in the NW and maximum foehn-induced melt in the NE (Fig. 5c). This time lag is longer for strong (>90th percentile) AR events and likely reflects the time required for the moisture plume to ascend the windward slope of the ice sheet, descend the leeward slope, and erode the low elevation cold pool. Finally, Fig. 5d shows that nearly all (75–100%) of extreme summer melt events (>99th percentile) occur during foehn in the ablation zone of the lower NEGIS catchment south to Storstrømmen glacier. Foehn contributions to extreme melt are also high (>50%) throughout the remainder of the NE Greenland ablation zone, including the eastern slopes of outlying ice caps, as well as in the middle elevations (1000–1500 m) of the NEGIS catchment.

Although these results are dependent on RACMO2 model simulations of ice sheet melt, the findings are similar when using model output from the Modèle Atmosphérique Régional (MAR) (Supplementary Fig. S5). MAR shows significantly higher foehn contributions to melt than RACMO2 at higher elevations due to its greater foehn frequency and less melt at high elevations, but in the low-elevation areas which contribute the vast majority (~95%) of melt, RACMO2 and MAR agree closely (Supplementary Fig. S6). Both the RACMO2 and MAR melt outputs compare well with satellite passive microwave melt data in NE Greenland (see Methods and Supplementary Information).

## Increasing atmospheric rivers driving extreme melt during the past two decades

We find a significant increase in the frequency of intense ($AR_{90}$) events in NW Greenland during 1980–2020 (Fig. 6a), with exceptionally frequent $AR_{90}$ occurrence during the record melt season of 2012[27]. The Mattingly and Wille algorithms detect ARs at different rates; the Wille method is optimized to exclude weaker events, and thus the frequency of ARs is more closely matched between the two algorithms when examining strong and extreme ARs compared to ARs of any intensity (Supplementary Fig. S7). Both methods show that extreme ARs (>99th percentile) have occurred almost exclusively after 2000 (Supplementary Fig. S7b).

To quantify the influence of AR and foehn events on the increasing trend in NE Greenland melt, we partition seasonally-integrated melt from RACMO2 into the four possible combinations of $AR_{90}$/no $AR_{90}$ and foehn/no foehn conditions (Fig. 6c–e). We fit a trend line to melt values during 1980–2020 using the Ensemble Empirical Mode Decomposition (EEMD) method, and calculate correlations between detrended melt anomalies and the detrended North Atlantic Oscillation (NAO) index (Fig. 6b) to assess the influence of regional atmospheric circulation changes on AR- and foehn-induced melt.

The most frequent melt condition is "no $AR_{90}$ or foehn", occurring 78.95% of the time (Fig. 6c). This condition accounted for the majority of total melt during every JJA season except 2006 due to its high frequency, but this category contributes relatively less to total melt compared to its proportional occurrence (Fig. 7). Melt not associated with $AR_{90}$ or foehn has increased linearly since the early 1990s and shows a strong negative correlation ($r = -0.51$) with the NAO index. Melt occurring with foehn and no $AR_{90}$ is the second most common category (14.97%) and is characterized by a step function-like increase around 2000, with several JJA seasons after 2000 having a large percentage of melt attributed to foehn (Fig. 6e). Foehn conditions produce melt more efficiently than during times with no $AR_{90}$ or foehn present, as the proportion of melt attributed to the "no $AR_{90}$ with foehn" category exceeds its frequency in nearly every season (Fig. 7). The long-term trend toward more negative NAO conditions has also influenced "no $AR_{90}$ with foehn" melt, with a correlation of −0.38. This suggests that, in addition to the importance of the AR-foehn connection, other atmospheric mechanisms (such as anticyclones lacking a corresponding AR[4]) can generate foehn-induced melt in NE Greenland.

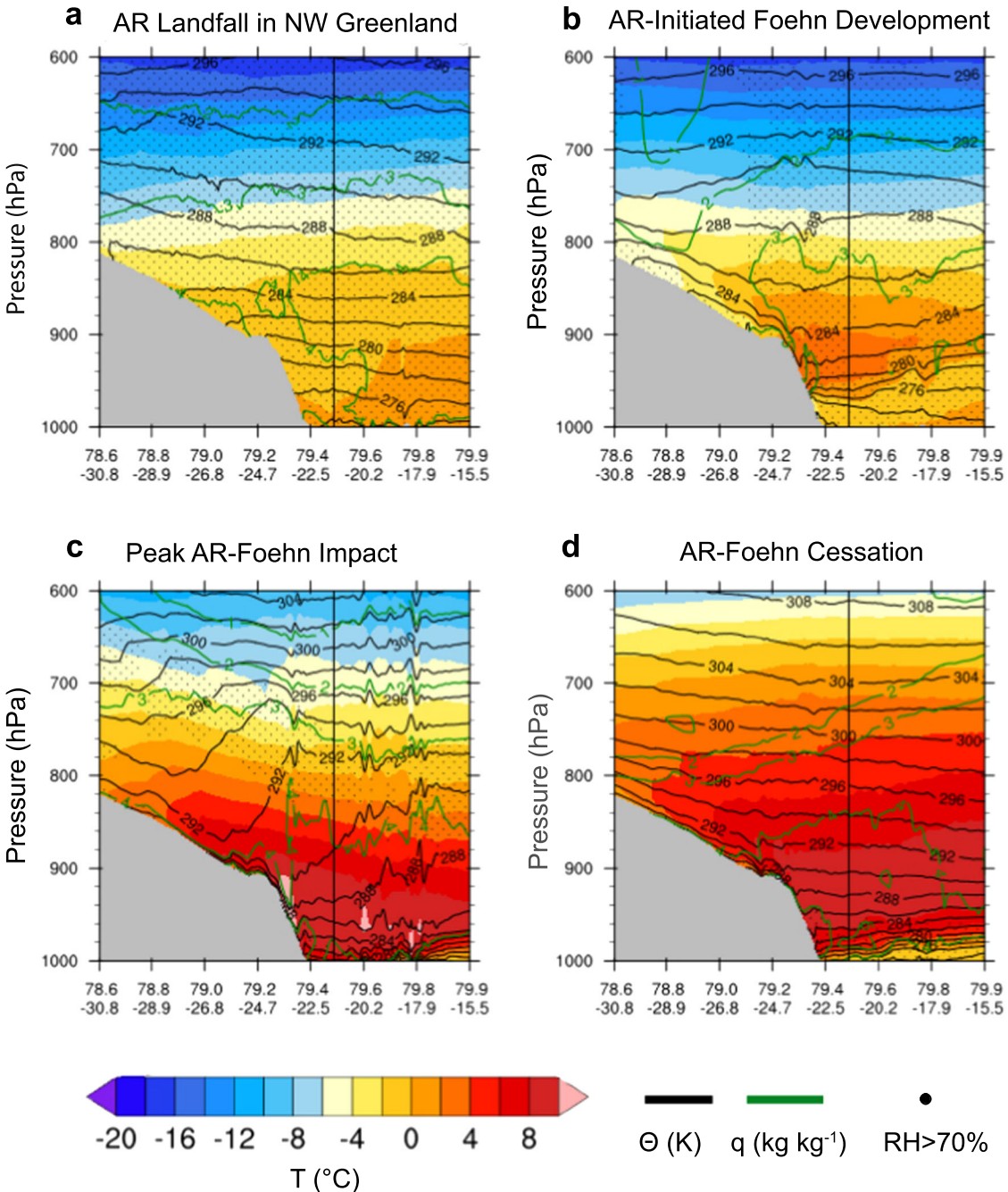

**Fig. 4 | Atmospheric cross sections of the July 2014 atmospheric river (AR)-foehn event over the 79°N Glacier region of northeast (NE) Greenland.** Cross sections are derived from Polar Weather Research and Forecasting (Polar WRF) model simulations and cover the transect shown by the black dashed line in Fig. 1b. The colored contours represent air temperature ($T$), black lines show the potential temperature ($\theta$), and green contours are the specific humidity ($q$). Relative humidity (RH) values above 70% are depicted by dots. The vertical black line highlights the location of the floating tongue of 79°N Glacier. The panels are during different time periods throughout the case study from (**a**) 18 July 2014 1200 UTC (AR landfall in NW Greenland); (**b**) 19 July 2014 1200 UTC (AR-initiated foehn development); (**c**) 20 July 2014 1200 UTC (Peak AR-foehn impact); and (**d**) 22 July 2014 1200 UTC (AR-foehn cessation).

Due to increasing $AR_{90}$ events (Fig. 6a), the amount of melt attributed to $AR_{90}$ conditions increased sharply after 2000. A steady linear increase in "$AR_{90}$ with foehn" melt has occurred since 2000, while non-foehn melt during $AR_{90}$ conditions has shown a slight decreasing trend since peaking in the extreme melt seasons of the early 2010s. $AR_{90}$ conditions contribute a highly disproportionate amount to total melt relative to their frequency, particularly when accompanied by foehn (Fig. 7). For example, during a few post-2000 seasons the proportion of melt attributed to the two $AR_{90}$ categories was 30–40%, despite less than 10% frequency of these conditions.

$AR_{90}$ + foehn conditions occur only 1.11% of the time, but these conditions account for around 5–10% of melt in several recent melt seasons. The effect of strong ARs on increasing melt trends is especially pronounced at higher elevations, where the total melt generated during $AR_{90}$ conditions at elevations above 1000 m is nearly as much as total melt below 1000 m during several seasons after 2000 (Supplementary Fig. S8). The NAO influence on the $AR_{90}$ melt categories is much less pronounced than on non-$AR_{90}$ categories, with a correlation of −0.09 with the "$AR_{90}$ with no foehn" category (Fig. 6d) and −0.24 with "$AR_{90}$ with foehn" (Fig. 6f).

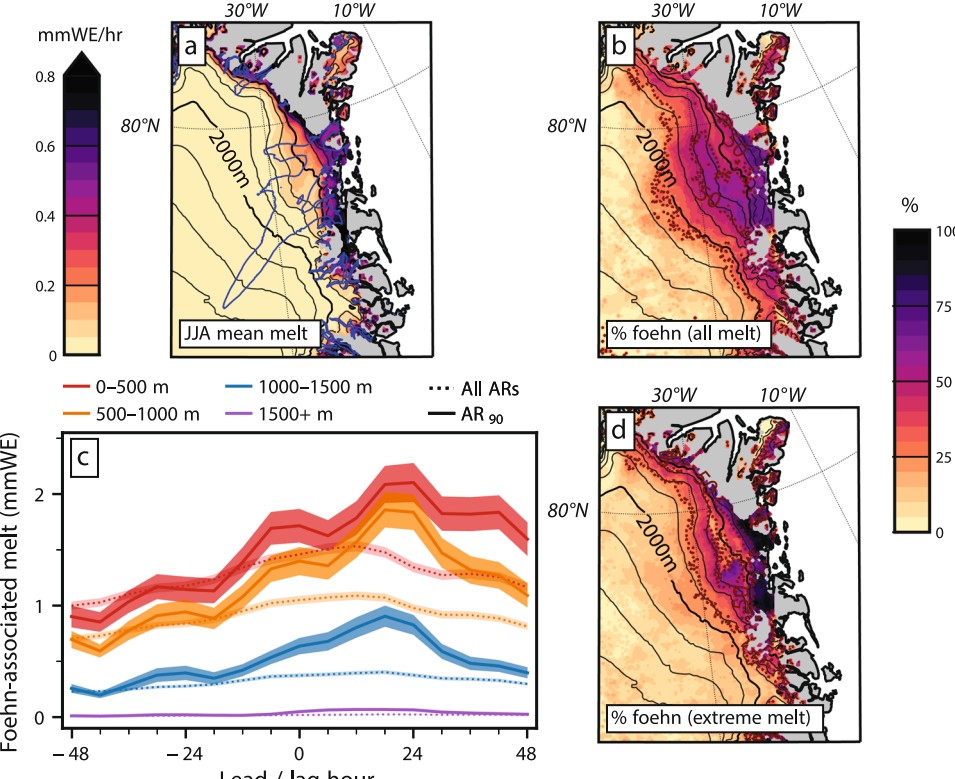

**Fig. 5 | Influence of atmospheric rivers (ARs) and foehn conditions on northeast (NE) Greenland melt. a** Climatological mean hourly melt in NE Greenland during summer (JJA). **b** Percentage of melt coincident with foehn conditions during the 0−48 h period after 90th percentile ARs (AR$_{90}$) in northwest (NW) Greenland. **c** Temporal evolution of foehn-driven melt in northeast (NE) Greenland in 500 m elevation bands during the −48 to +48 h period surrounding NW Greenland ARs. Lines are mean values and shading displays the standard error of the mean. (**d**) Map of percentage of extreme (>99th percentile) melt coincident with foehn conditions. Melt data are from the polar Regional Atmospheric Climate Model version 2.3p2 (RACMO2) and ARs are identified from the Modern-Era Retrospective analysis for Research and Applications, Version 2 (MERRA-2) reanalysis using the Mattingly algorithm.

## Discussion

Since the beginning of the 21st century, northern Greenland has entered a new paradigm with NW Greenland ARs becoming more frequent and reaching higher intensity thresholds, leading to significant increases in extreme foehn-driven melt events in the NE. Supraglacial lakes now develop at higher elevations and persist longer after the melt season ends[13], and are projected to expand further inland under the effects of climate change[35,36], coinciding with velocity increases for the glaciers comprising the NEGIS. The connection between increased surface melt and faster glacial flow is supported by an ice sheet model sensitivity study demonstrating that ice flow in the NEGIS and its outlet glaciers is highly sensitive to changes in surface mass balance[37], particularly in the low-elevation areas of NE Greenland where we find large AR$_{90}$ and foehn contributions to extreme melt. Ice flow in this basin is also influenced by the 80 km × 20 km floating tongue of 79°N Glacier, which is exposed to both a warming atmosphere and warming ocean. While this ice shelf remains intact, the grounding line of the glacier has retreated by 4.4 km[10] and there has been a pronounced thinning of 1 m yr$^{-1}$ since 2009[9]. Similar to processes observed over the Antarctic Peninsula, basal melt preconditions and reduces ice shelf stability, while widespread melt ponds resulting from AR-induced foehn winds could trigger a break-up of the 79°N Glacier's ice tongue.

Our findings show that future changes in NE Greenland extreme melt events, the NEGIS, and consequently the sea level rise contribution from NE Greenland will strongly depend on the interaction between Arctic warming and moistening, and the evolution of regional atmospheric circulation patterns. ARs are often dynamically linked with blocking anticyclones[38], which are termed "Greenland blocks" when they occur in the vicinity of Greenland[39]. Future trends in atmospheric dynamics, including ARs and blocking, over Greenland and the broader North Atlantic region are highly uncertain. CMIP5 and CMIP6 model simulations do not project any increasing trend in Greenland blocking in future warming scenarios, but these models failed to capture the increases in Greenland blocking that have contributed to enhanced Greenland melt in recent decades[40,41] and underestimate blocking frequency in the North Atlantic region more generally[42]. Notably, our results show that extreme melt events induced by the AR-foehn mechanism are only weakly correlated with the NAO, suggesting that the AR influence on NE Greenland extreme melt may continue to increase even if recent negative NAO conditions do not persist. Climate model simulations project greater AR intensity due to increased atmospheric moisture content[43,44], and poleward moisture transport into the Arctic is expected to increase with climate warming[45]. Additionally, the decline of local sea ice cover will likely be a unique regional factor contributing to increased moisture availability and changes in atmospheric circulation in northern Greenland[40,46].

Taken together, the high confidence in increased atmospheric moisture content, combined with uncertain projections of large-scale circulation patterns, lead us to conclude that the AR-induced foehn melt mechanism will likely continue to increase in influence on NE Greenland mass loss. Even relatively infrequent extreme warm season AR events, strengthened by increased future atmospheric moisture content, will have long-lasting and compounding impacts on runoff, solid ice flow, and firn structure that will enhance NE Greenland's contribution to sea level rise.

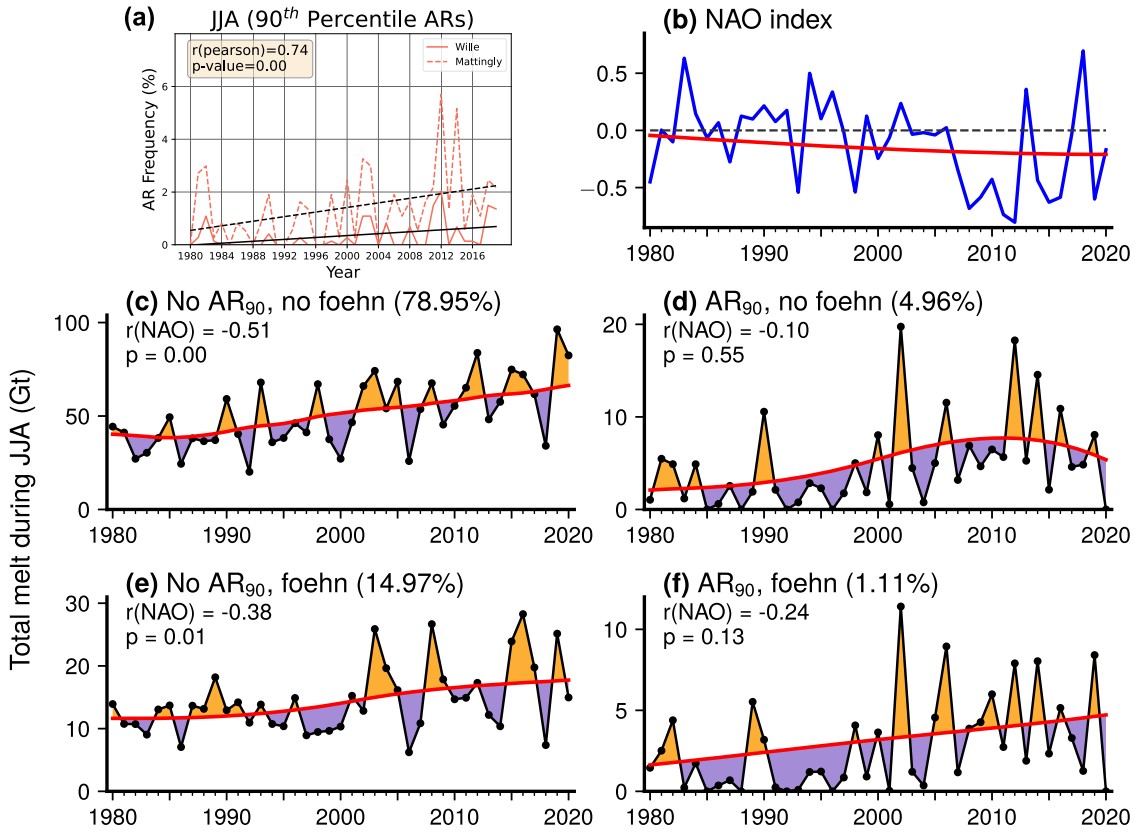

**Fig. 6 | Northern Greenland summer (JJA) atmospheric river (AR) and foehn time series. a** JJA AR frequency trends in northwest (NW) Greenland (orange box in Fig. 1a) for ARs exceeding the 90th intensity percentile (AR$_{90}$) (from monthly climatological integrated water vapor transport (IVT) during 1980–2020) according to the AR detection algorithms in refs. [21,31], with linear trend fit superimposed. **b** JJA mean North Atlantic Oscillation (NAO) index during 1980–2020, with nonlinear curve fit using the Ensemble Empirical Mode Decomposition (EEMD) method (red line). Time series of total JJA melt (Gt) from the polar Regional Atmospheric Climate Model version 2.3p2 (RACMO2) integrated across all grid cells in northeast (NE)

Greenland domain, partitioned into the four possible combinations of AR$_{90}$ and foehn conditions: (**c**) no AR$_{90}$ with no foehn; (**d**) AR$_{90}$ with no foehn; (**e**) no AR$_{90}$ with foehn; (**f**) AR$_{90}$ with foehn. Red line shows EEMD curve fit and filled areas are positive and negative anomalies with the EEMD trend removed, which are used to calculate correlations with the detrended NAO index annotated on plots. The numbers in parentheses are the overall frequency of the given AR$_{90}$ + foehn combination of conditions during JJA 1980–2020. Note that melt is attributed to AR$_{90}$ conditions anytime a > 90th percentile AR was detected in the NW Greenland domain within the prior 48 h.

## Methods

### AR detection

Two different AR detection algorithms are employed in this study for comparison purposes. The first algorithm was developed by ref. [27] and is applied to fields of integrated water vapor transport (IVT) calculated from 6-h Modern-Era Retrospective analysis for Research and Applications, Version 2 (MERRA-2) reanalysis data, which has 0.5° × 0.625° horizontal resolution[47]. IVT (in units of kg m$^{-1}$ s$^{-1}$) is calculated using the formula

$$IVT = \frac{1}{g} \int_{1000hPa}^{200hPa} q\mathbf{V}\,dp \qquad (1)$$

where $g$ (m s$^{-2}$) is the gravitational acceleration, $q$ (kg kg$^{-1}$), and $\mathbf{V}$ (m s$^{-1}$) are the specific humidity and vector wind, respectively, at a given atmospheric pressure level, and $dp$ is the increment between pressure levels ($dp$ = 50 hPa between 1000 and 500 hPa; $dp$ = 100 hPa between 500 and 200 hPa). The algorithm first detects areas where IVT is greater than a minimum threshold of 150 kg m$^{-1}$ s$^{-1}$ and exceeds the 85th climatological percentile, where climatology is defined by the distribution of all values during 1980–2016 within a 31-day centered window at a given grid point. Areas where these conditions are met are then filtered by several size, shape, and location criteria to ensure the final AR record is composed of long, narrow moisture transport features in the Northern Hemisphere mid-latitude and polar regions.

The algorithm also requires moisture transport to be in the poleward direction for mid-latitude ARs, but relaxes this requirement for ARs north of 70°N in order to detect moisture plumes originating in the Arctic that may affect Greenland. We tested a version of this detection algorithm forced with ERA5 reanalysis and found a negligible difference in AR detection frequency and AR-related melting. Further details on this AR detection method can be found in refs. [21,27].

The other AR detection method utilized in this study is a polar-specialized algorithm originally developed for use in the Antarctic by refs. [25,31]. This algorithm identifies grid cells between 42.5° and 85°N for values of vIVT (the meridional ($v$) component of IVT) within the 98th percentile of all monthly vIVT values from 1980 to 2020. If a filament of vIVT values within this percentile extends at least 20° in the meridional direction, it is identified as an AR. The vIVT (kg m$^{-1}$ s$^{-1}$) term was calculated as

$$vIVT = -\frac{1}{g} \int_{surface}^{top} qv\,dp \qquad (2)$$

where $q$, $g$, and $dp$ are the same as described above, and $v$ is the meridional component of wind velocity (m s$^{-1}$). Like other AR detection algorithms in the Atmospheric River Tracking Method Intercomparison Project (ARTMIP), this algorithm is applied to 3-hourly fields on all reanalysis levels from MERRA-2. For additional details regarding this AR detection algorithm, see ref. [31].

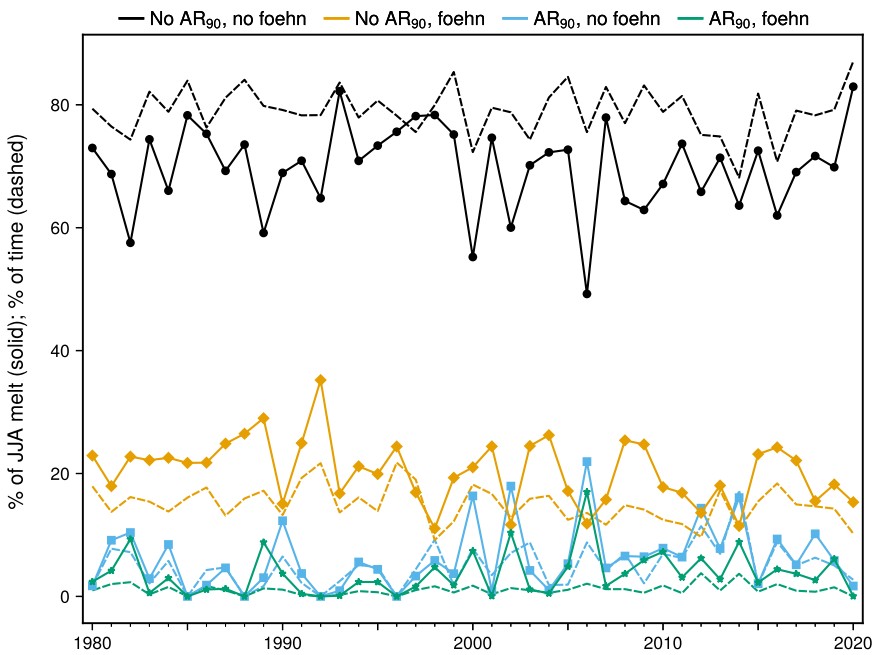

**Fig. 7 | Time series of northeast (NE) Greenland summer (JJA) melt attributable to combined atmospheric river (AR) and foehn conditions.** Plotted is polar Regional Atmospheric Climate Model version 2.3p2 (RACMO2) simulated cumulative JJA melt (solid line) attributed to the four possible combinations of 90th percentile atmospheric river (AR$_{90}$) and foehn conditions: no AR$_{90}$ with no foehn; no AR$_{90}$ with foehn; AR$_{90}$ with no foehn; AR$_{90}$ with foehn. Also plotted is the percentage of time during each JJA season classified into each category (dashed lines).

## RACMO2 and MAR regional climate models

RACMO2 is specifically adapted to represent surface processes in polar regions[48]. Here RACMO2 is run at 5.5 km spatial resolution over a domain including the GrIS, its peripheral ice caps, and glaciers of the northern Canadian Arctic, Svalbard and Iceland[3]. The model incorporates the dynamical core of the High Resolution Limited Area Model (HIRLAM) and the physics package of the European Centre for Medium-Range Weather Forecasts-Integrated Forecast System (ECMWF-IFS). RACMO2 includes a 40-layer snow module (up to 100 m depth) simulating melt, percolation and retention into firn, and runoff. It represents dry-snow densification, drifting snow erosion, and snow albedo based on grain size, cloud optical thickness, solar zenith angle, and impurity content (soot).

RACMO2 is forced by a combination of climate reanalyses including ERA-40 (1958–1979), ERA-Interim (1980–1989), and ERA5 (1990–2021) within a 24-grid-cell-wide relaxation zone at the model lateral boundaries. Upper atmospheric relaxation is also active. Forcing consists of temperature, pressure, specific humidity, wind speed and direction being prescribed at the 40 model atmospheric levels every 6 h (ERA-40 and ERA-Interim) or 3 h (ERA5). Sea ice extent and sea surface temperature are also prescribed from the ERA reanalyses on a 6-hourly or 3-hourly basis. Firn is initialized in September 1957 using snow temperature and density profiles from the offline Institute for Marine and Atmospheric research Utrecht-Firn Densification Model (IMAU-FDM). Ice albedo is prescribed from the 500 m Moderate Resolution Imaging Spectroradiometer (MODIS) 16-day surface albedo product (MCD43A3) as the lowest 5% annual values averaged for the period 2000–2015, ranging from 0.30 for dark bare ice to 0.55 for clean bright ice under perennial firn. For detailed model description and evaluation using in situ and remote sensing measurements, we refer to refs. [3,48].

MAR is a regional climate model especially developed for studying the near surface climate and SMB of both polar ice sheets where MAR has been widely validated with in situ observations and satellite data sets. With RACMO2, it is notably one of the best models currently available for simulating the GrIS SMB[49]. The version 3.12.0 of MAR has been run here at a resolution of 15 km, 1-hourly forced by the ERA5

reanalysis since 1950. With respect to MARv3.11 (fully described in ref. [50]), the main improvements of MARv3.12 are the geographical projection used by MAR which is now the standard Polar Stereographic EPSG 3413, a correction of an important bug impacting the snow temperature at the base of the snowpack, a conservation of water mass in the soil impacting water fluxes over the tundra, and a continuous conversion from rainfall to snowfall from 0 °C to −2 °C as input of the snow model instead as a fixed one at −1 °C.

MAR ice mask is a percentage value, so the foehn algorithm is only applied where permanent ice covers >10% of a grid cell. In RACMO2, the ice mask is binary, with SMB components only calculated for grid cells with ice coverage ≥50%. We use the version of the RACMO2 mask which includes peripheral ice caps, so that we can include Flade Issblink in our analysis.

## Comparison of RACMO2 and MAR melt area with satellite passive microwave melt data

To evaluate the RACMO2 and MAR melt simulations, we used the NASA MEaSUREs Greenland Surface Melt Daily 25 km EASE-Grid 2.0 dataset[51] during 1980–2020. This dataset classifies the ice sheet surface into melt/no melt areas with daily temporal resolution, based on brightness temperature data acquired by three passive microwave radiometers: the Scanning Multichannel Microwave Radiometer (SMMR), the Special Sensor Microwave/Imager (SSM/I), and the Special Sensor Microwave Imager/Sounder (SSMIS). This dataset has in turn been compared to Greenland station observations of melt conditions and compares favorably to other melt retrieval algorithms applied to passive microwave satellite data[52].

In order to compare the model and satellite data, we first bilinearly interpolated the RACMO2 and MAR melt data to the 25 km EASE Grid 2.0 on which the MEaSUREs melt data is provided, and masked out grid cells in the NE Greenland domain that are not classified as ice sheet by all three datasets. Since the passive microwave melt data is provided as binary melt/no melt values with daily temporal resolution, whereas RACMO2 (MAR) outputs meltwater production values with 3-hourly (hourly) resolution, we developed a method to compare the

model meltwater production values with the binary satellite values. We summed the RACMO2 & MAR meltwater production values for each day, then tested a range of threshold values of meltwater production (mmWE/day) for classification as a "melt day" in the model datasets. Previous studies have used a threshold of 1 mmWE/day[53] or 8.25 mmWE/day[54,55] to classify "melt days" using MAR, so we tested the following meltwater production thresholds: 0.1, 1, 2, 5, 8.25, 10 mmWE/day.

Using all available values from ice sheet grid cells in the NE Greenland domain during JJA 1980–2020, we produced a confusion matrix for each model melt threshold. The confusion matrix classifies each pixel on each day into one of four categories: "true negative" (model = no melt, MEaSUREs = no melt); "false positive" (model = melt, MEaSUREs = no melt); "false negative" (model = no melt, MEaSUREs = melt); and "true positive" (model = melt; MEaSUREs = melt). Then we calculated the following summary statistics from the confusion matrix at each threshold:

- Accuracy: (TP + TN)/total
- True positive rate: TP/actual yes
- False positive rate: FP/actual no
- True negative rate: TN/actual no

The validation summary statistics are provided in Supplementary Table S1. They show that a threshold of 2 or 5 mmWE/day provides the highest accuracy of the thresholds tested for both RACMO2 and MAR, with lesser accuracy for lower and higher thresholds. Since the overall prevalence of melt is relatively low (10.21%) across all grid cells, a high true positive rate is important to demonstrate that the models can capture melt when it actually does occur, while maintaining a low false positive rate. Because the true positive rate for the 2 mmWE/day threshold is ~12–15% higher than the 5 mmWE/day threshold, with a false positive rate only ~1–2% higher, we consider 2 mmWE/day the optimal threshold value for classification of a "melt day" in the models in NE Greenland.

Finally, to evaluate how well RACMO2 and MAR simulate the spatial pattern of NE Greenland melt, we created maps of the overall percentage of JJA days with melt for each dataset and their differences with the MEaSUREs data. The plots for the 1, 2, and 5 mmWE/day thresholds are shown in Supplementary Figs. S9–S11. These maps show that RACMO2 and MAR broadly agree with MEaSUREs on the spatial pattern of mean JJA melt. RACMO2 overestimates the frequency of melt in most areas with a 1 mmWE/day threshold and underestimates melt prevalence with a 5 mmWE/day threshold, with a relatively even spatial distribution of overestimation and underestimation for the 2 mmWE/day threshold. MAR has a more definite spatial pattern to its bias, with an underestimation of melt frequency at higher elevations even for the 1 mmWE/day threshold, and an overestimation of melt frequency at lower elevations even for the 5 mmWE/day threshold. These results show that the models generally simulate the melt area in NE Greenland accurately when an appropriate threshold is found for comparison with independent passive microwave satellite data, and that RACMO2 is more consistent than MAR in its agreement with the satellite melt data across elevation ranges.

### Foehn Identification Algorithm

The classification of foehn events is non-trivial, and there is no established best practice[29,56]. Assumptions are made that foehn has a sufficiently distinct signature at a given location to be identified using (1) thresholds of key meteorological fields (e.g., ref. [29]), (2) changes in conditions from pre- and post-foehn conditions (e.g., ref. [57]) or (3) a combination of both (e.g., ref. [24]) to capture the foehn onset and development, as well as their features throughout the event. No Foehn-Identification Algorithm (FIA) previously existed for Greenland, therefore the following method is developed for detection of foehn in MAR and RACMO2.

For grid cells where the ice mask threshold is greater than 10%, foehn is detected from six-hourly averaged values when the following conditions are simulated:

- Wind direction between 220° and 350° AND
- Wind speed greater than 5 m s$^{-1}$ AND
- A relative humidity value less than the 15th percentile of a 2-week window surrounding the given date OR
- A 5% decrease in relative humidity AND 3 °C increase in temperature compared to the previous 6-h value

The westerly wind requirement removes the possibility of including warm air flow from the Atlantic with passing storms. The combination of relative humidity decrease and simultaneous temperature increase reduces the possibility of including cold katabatic winds in the results, and follows similar FIAs in the Antarctic (e.g., refs. [24,57]). The 15th percentile for relative humidity is chosen based on our exploratory case study analysis of foehn events in both AWS data from KPC_L and KPC_U, and MAR data (not shown). This threshold detects most foehn cases whilst excluding other westerly flows such as katabatic winds. Note that this FIA is developed specifically for NE Greenland and would require modification for other areas of Greenland where downslope winds are not westerly in direction.

Several sensitivity studies were run to test variations on the FIA, including varying the temperature and relative humidity change, adopting a 10% relative humidity threshold, and varying the wind speed requirement, all of which led to small changes in the number of foehn grid cells identified, but did not alter the overall results and conclusions of the study.

### Polar Weather Research and Forecasting model simulation

The high-resolution case study simulation using the Polar Weather Research and Forecasting (Polar WRF) model was previously simulated and evaluated by ref. [32] and was shown to sufficiently represent the air temperature and wind components throughout the year. The specific humidity in summer was overestimated significantly. However, as we use this simulation purely as a case study this is not found to influence the results or conclusions.

### Ensemble Empirical Mode Decomposition (EEMD) time series analysis

The time series of NAO, Greenland Blocking Index (GBI), and melt attributed to the four possible combinations of AR$_{90}$ and foehn conditions are fit with trend lines using the nonlinear Ensemble Empirical Mode Decomposition (EEMD) method (ref. [58]) using the Rlibeemd package for the R programming language[59]. Correlations between anomalies in NAO/GBI and melt values are calculated by subtracting the trend fit from each time series. We present only the NAO results in the paper, as the melt time series correlations with GBI are qualitatively similar but with a reversed sign, and we also note that correlations are similar using the actual (non-detrended) values of each variable.

### Data availability

RACMO2 data were published in ref. [3] and additional variables, including 5.5 km resolution melt output, are available upon request from Brice Noël (b.p.y.noel@uu.nl) or Michael van den Broeke (M.R.vandenBroeke@uu.nl). MAR data[60] are available at https://doi.org/10.5281/zenodo.7591112. Passive microwave GrIS daily melt data[51] are available at https://doi.org/10.5281/zenodo.7591560. Output from the Mattingly and Wille AR detection algorithms can be downloaded from the ARTMIP[61] Tier 1 database at https://doi.org/10.5065/D62R3QFS. The Polar WRF model output[62] for 2014–2018 is available at https://doi.org/10.17605/OSF.IO/53E6Z. ERA5[63] data are available from the Copernicus Climate Change Service Climate Data Store: https://cds.climate.copernicus.eu/#!/search?text=ERA5&type=dataset. Daily time series of the NAO index are obtained from the NOAA

National Weather Service Climate Prediction Center: https://www.cpc.ncep.noaa.gov/products/precip/CWlink/pna/nao.shtml. Daily time series of the GBI calculated using NCEP / NCAR reanalysis data are obtained from the NOAA Earth System Research Laboratories Physical Sciences Laboratory: https://psl.noaa.gov/data/timeseries/daily/GBI/. Near-surface meteorological observations from the Programme for Monitoring of the Greenland Ice Sheet (PROMICE) dataset[64] are available at https://www.promice.org/PromiceDataPortal/#Automatic weatherstations. The annual mean GrIS velocity is obtained from the MEaSUREs Multi-year Greenland Ice Sheet Velocity Mosaic, Version 1 dataset[65] distributed by the National Snow and Ice Data Center (NSIDC): https://nsidc.org/data/NSIDC-0670/versions/1. MODIS imagery from the NASA Worldview application, part of the NASA Earth Observing System Data and Information System (EOSDIS), can be obtained here: https://worldview.earthdata.nasa.gov.

## Code availability
All code used for analysis and figure generation can be obtained from the authors upon request.

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

## Acknowledgements

K.S.M. acknowledges support from the Polar Radiant Energy in the Far InfraRed Experiment (PREFIRE) mission, NASA grant 80NSSC18K1485. J.D.W. acknowledges support from the Agence Nationale de la Recherche project, ANR-20-CE01-0013 (ARCA). B.N. was funded by the NWO VENI grant VI.Veni.192.019.

## Author contributions

K.S.M., J.V.T., and J.D.W. formulated the study. J.V.T. developed the foehn identification algorithm and analyzed the WRF model simulations. K.S.M., J.V.T., and J.D.W. analyzed the influence of ARs and foehn on ice sheet melt. B.N. and X.F. contributed to the analysis and interpretation of the RACMO2 and MAR regional climate model outputs. Å.K.R. and T.L.M. contributed to the analysis and interpretation of the passive microwave satellite melt data. K.S.M., J.V.T., and J.D.W. led the writing with significant input from all authors.

## Competing interests

The authors declare no competing interests.
