## [Peer Review File · Nature Communications]

Increasing extreme melt in northeast Greenland linked to foehn winds and atmospheric riversREVIEWER COMMENTS

Reviewer #1 (Remarks to the Author):

General Comments:

The authors have presented some nice analysis. However, one fundamental aspect is whether the extreme melting events are an important enough contributor to surface melting in northeast Greenland, and thus to sea level rise, to justify publication in Nature Communications. The focus in the manuscript is on foehns and atmospheric rivers. Figure 2 shows the percentage of melt occurring with atmospheric rivers (AR) to be most important at high elevations and much less so at lower elevations where the majority of melting occurs. Looking at Figure 6 to get quantitative results, the combination of AR90 plus foehns contributes only about 5% of the total melt from RACMO output, eyeballing the average melt since 2000. (What are the percentages in parentheses next to the titles?). Another way of looking at this is to examine Figure ED8 and taking similar averages for 500-1000 m elevation band where the melting is most developed, and one reaches a similar conclusion as to the importance of AR90 plus foehns. Yes, there are large spikes but there are also low years. Even if this combination was to double in impact during the coming decades, we are only talking about 10% of the total, assuming other components don't significantly change. Realistically though, the combination of no AR90 and no foehns dominates the melting now and for the near future, roughly 2/3rds of the total. What will the future trajectory of this combination and the NAO?

The second major area of concern is that the authors are dealing with only model output. The results from RACMO in Figure 5 in panels b and d (melt with foehn) don't look at all similar to the same panels from MAR shown in Figure ED5. The marked divergence of the model results implies the results are not nearly as robust as claimed, and this is for the regions where the differing model resolutions (5.5 km versus 15 km) should be much less important. The authors don't help themselves having different vertical scales for panel c in the two figures. Appeal to independent observations of ice surface melt, like from satellite passive microwave or scatterometry, would significantly strengthen the claims being made especially for the model output of extreme melting and should be provided. This is not something that should be relegated to material published elsewhere.

My recommendation is to provide the authors with the opportunity to make a much stronger case for the combined importance of atmospheric rivers and foehn events.

Specific Comments:

- 1. Does one get different results for the ARs and their extremes if ERA5 rather than MERRA2 is used for the integrated vapor transport?**
- 2. Figure 5 needs NEGIS like Figure ED5.**
- 3. Figure ED2: What is the horizontal scale of the MODIS imagery? I am not sure what is bare ice and melt ponds in the pictures.**
- 4. Figure ED3 (and ED2): Aren't the shortwave anomalies due to foehn clearance on the lee side important?**
- 5. Figure ED4: It is surprising that the relative humidity didn't drop more sharply. Often foehns are very dry on the lee side.**
- 6. Figure ED10: I didn't get much out of this figure to support the case being made. Why use melt from MAR here?**

Reviewer #2 (Remarks to the Author):

General: This is a well-written paper and quite complete. I do have a few questions:

1) Figure 3 shows increasing melt with higher IVT but also suggests that there is still some melt even at low IVT. Presumably there is some melt at the edges of the ice sheet at the height of summer simply because it gets warm. It would be interesting to look at the relationship in the wings of the summer season.

2) The case in Figure1 shows an AR originating over Canada. This seems a little rare when looking at Hermann et al. (ref below) where they show back trajectories mostly from the south and not that many linked to melting in the NE.

3) Are there ever Foehn conditions to the east without an AR in the NW?

Specific:

Line 113: Reference is to Basin 8 – This is the only reference to a basin number?

Lines 124-127: Sometimes, in cases of a mountain range with a flat plane to the east, lee waves associated with the foehn effect can lower the surface pressure at the base of the mountain range drawing cooler air from the east towards the base of the mountains. A higher resolution model might show this.

Line 381: This link does not work:

<https://www.earthsystemgrid.org/dataset/ucar.cgd.cesm4.artmip.tier1.html>.

Figures:

Figure 2: The color scheme here leaves something to be desired. Perhaps putting white contour lines at, say 50% and 75%, would make it easier to see differences in the results.

Reference:

Hermann, M., L. Papritz, and H. Wernli (2020), A Lagrangian analysis of the dynamical and thermodynamic drivers of large-scale Greenland melt events during 1979–2017, *Weather Clim. Dynam.*, 1(2), 497-518, doi:10.5194/wcd-1-497-2020.

Reviewer #1 (Remarks to the Author):

General Comments:

The authors have presented some nice analysis. However, one fundamental aspect is whether the extreme melting events are an important enough contributor to surface melting in northeast Greenland, and thus to sea level rise, to justify publication in Nature Communications. The focus in the manuscript is on foehns and atmospheric rivers. Figure 2 shows the percentage of melt occurring with atmospheric rivers (AR) to be most important at high elevations and much less so at lower elevations where the majority of melting occurs. Looking at Figure 6 to get quantitative results, the combination of AR90 plus foehns contributes only about 5% of the total melt from RACMO output, eyeballing the average melt since 2000. (What are the percentages in parentheses next to the titles?). Another way of looking at this is to examine Figure ED8 and taking similar averages for 500-1000 m elevation band where the melting is most developed, and one reaches a similar conclusion as to the importance of AR90 plus foehns. Yes, there are large spikes but there are also low years. Even if this combination was to double in impact during the coming decades, we are only talking about 10% of the total, assuming other components don't significantly change. Realistically though, the combination of no AR90 and no foehns dominates the melting now and for the near future, roughly 2/3rds of the total. What will the future trajectory of this combination and the NAO?

Our primary focus in this study is on the contribution of ARs and foehn to extreme melt events because of the disproportionate impact of these extreme events on overall ice sheet evolution. Due to their infrequency, the majority of cumulative NE Greenland surface melt does not directly correspond in time to these events, and this will likely continue to be the case even in high-end climate change scenarios. However, our results show that a highly disproportionate amount of melt occurs during these events – for example, despite AR90 + foehn conditions occurring only 1.11% of the time, Fig. ED10 shows that these conditions account for around 5–10% of melt in several recent melt seasons. On the other hand, no AR90 + no foehn conditions prevail most of the time, but account for a disproportionately small amount of melt in relation to their frequency.

Further, as we note in the paper (with a few minor edits in the revised version to make this point more clearly), extreme melt events have short- and long-term compounding impacts that can't be measured by calculating only the total amount of melt immediately produced during these short-lived events. The sudden introduction of huge amounts of meltwater alters the ice sheet albedo, firn structure, and supra- and sub-glacial hydrology in ways that are fundamentally different from producing a larger cumulative amount of melt more gradually over the course of a melt season. Depending on their seasonal timing, extreme melt events can lower the ice sheet albedo for the remainder of the melt season and prime the surface for greater melt during “normal”

weather conditions. They can overwhelm the subglacial drainage system with a meltwater pulse and lead to faster solid ice flow. On longer time scales, the impact of an extreme melt event taking place over the course of only a few days can persist for years in the form of ice slabs that prevent meltwater from percolating into firn and enhance runoff. As noted in the paper, these ice slabs are prevalent in NE Greenland.

With regard to the future trajectory of the no AR90 + no foehn combination and the NAO, we note in the Discussion section of the paper that the future trajectory of the NAO is highly uncertain due to the issues that climate models have with simulating NAO variability in the historical record. Given the strong inverse relationship between no AR90 + no foehn melt and the NAO, we expect that future melt during these conditions will be closely linked to the future evolution of the NAO.

The percentages in Fig. 6 are the overall frequency of the given AR90 + foehn combination of conditions during JJA 1980–2020. This information has been added to the figure caption – thanks to the reviewer for pointing this out.

The second major area of concern is that the authors are dealing with only model output. The results from RACMO in Figure 5 in panels b and d (melt with foehn) don't look at all similar to the same panels from MAR shown in Figure ED5. The marked divergence of the model results implies the results are not nearly as robust as claimed, and this is for the regions where the differing model resolutions (5.5 km versus 15 km) should be much less important. The authors don't help themselves having different vertical scales for panel c in the two figures. Appeal to independent observations of ice surface melt, like from satellite passive microwave or scatterometry, would significantly strengthen the claims being made especially for the model output of extreme melting and should be provided. This is not something that should be relegated to material published elsewhere.

We have performed several new analyses to show that (1) RACMO and MAR agree well with each other on the amount of foehn-related melt in lower elevations (where the vast majority of melt occurs in NE Greenland); and (2) RACMO and MAR melt *area* agree well with satellite passive microwave data after selecting an appropriate meltwater production threshold for delineating melt area in the models. We originally chose to display RACMO results in the main paper because of its higher spatial resolution, which enables a more detailed picture of spatial variability in melt in the ablation zone, and the fact that our main results are not qualitatively dependent on the choice of model. These new analyses reinforce this decision by demonstrating that RACMO agrees better than MAR with the satellite data with regard to the spatial pattern of melt across all elevation zones. We have added the model comparisons with satellite passive microwave data to the extended data section of the paper.

First, the discrepancy in appearance between the RACMO (Figs. 5b/d) and MAR (Figs. ED6b/d) foehn-related melt maps primarily occurs at higher elevations, where MAR simulates a higher frequency of foehn and less melt than RACMO. The higher foehn frequency coupled with the less overall melt means that MAR attributes more melt to foehn at higher elevations during AR events.

Overall JJA foehn frequency from RACMO (left; reproduced from Fig. ED7a) and MAR (right).

Mean JJA hourly melt from RACMO (left) and MAR (right), with shifted color scale to highlight low values at higher elevations.

To compare the RACMO and MAR results in the areas of NE Greenland where substantial melt occurs, we masked out areas where the average cumulative melt during JJA is < 100 mmWE. Despite only covering 27.9% (32.9%) of the NE Greenland domain in RACMO (MAR), these

≥ 100 mmWE melt areas account for 93.6% (96.4%) of total melt produced during an average summer. Comparing Figs. 5b/d and Figs. ED6b/d with these areas masked out, we see that RACMO and MAR agree well on the magnitude and spatial pattern of foehn-related melt in the lower-elevation areas where the vast majority of meltwater in NE Greenland is produced.

Reproductions of Fig. 5b (left; RACMO) and ED6b (right; MAR). Areas with JJA mean cumulative melt < 100 mmWE are masked out.

Reproductions of Fig. 5d (left; RACMO) and ED6d (right; MAR). Areas with JJA mean cumulative melt < 100 mmWE are masked out.

Additionally, we changed the y-axis bounds to be the same for Fig. 5c and Fig. ED6c – thanks to the reviewer for pointing this out. These revised figures also show that MAR and RACMO agree well on the magnitude of foehn-related melt during AR events, except in the 1000–1500m elevation band where MAR shows substantially more melt than RACMO due to a greater number of grid cells – and higher melt in these grid cells – in the 1000–1500m band in the southeast part of the NE Greenland domain.

Reproductions of Fig. 5c (left; RACMO) and ED6c (right; MAR). The y-axis boundaries have been updated to be the same for both models.

To validate the RACMO and MAR melt simulations, we used the NASA MEaSUREs Greenland Surface Melt Daily 25km EASE-Grid 2.0 dataset (Mote, 2014) during 1980–2020, with the data for 2013–2020 provided by coauthor Mote. This dataset classifies the Greenland Ice Sheet surface into melt / no melt areas with daily temporal resolution, based on brightness temperature data acquired by three passive microwave radiometers: the Scanning Multichannel Microwave Radiometer (SMMR), the Special Sensor Microwave/Imager (SSM/I), and the Special Sensor Microwave Imager/Sounder (SSMIS). This dataset has in turn been compared to Greenland station observations of melt conditions and compares favorably to other melt retrieval algorithms applied to passive microwave satellite data (Kimball et al., 2021).

In order to compare the model and satellite data, we first bilinearly interpolated the RACMO and MAR melt data to the 25km EASE Grid 2.0 on which the MEaSUREs melt data is provided, and masked out grid cells in the NE Greenland domain that are not classified as ice sheet by all three datasets. Since the passive microwave melt data is provided as binary melt / no melt values with daily temporal resolution, whereas RACMO (MAR) outputs meltwater production values with 3-hourly (hourly) resolution, we developed a method to compare the model meltwater production values with the binary satellite values. We summed the RACMO & MAR meltwater production values for each day, then tested a range of threshold values of meltwater production (mmWE / day) for classification as a “melt day” in the model datasets. Previous studies have used a threshold of 1 mmWE / day (Mioduszewski et al. 2016) or 8.25 mmWE / day (Fettweis et al.

2011; Alexander et al. 2019) to classify “melt days” using MAR, so we tested the following meltwater production thresholds: 0.1, 1, 2, 5, 8.25, 10 mmWE / day.

Using all available values from ice sheet grid cells in the NE Greenland domain during JJA 1980–2020, we produced a confusion matrix for each model melt threshold. The confusion matrix classifies each pixel on each day into one of four categories: “true negative” (model = no melt, MEaSURES = no melt); “false positive” (model = melt, MEaSURES = no melt); “false negative” (model = no melt, MEaSURES = melt); and “true positive” (model = melt; MEaSURES = melt). Then we calculated the following summary statistics from the confusion matrix at each threshold:

- Accuracy: $(TP+TN) / \text{total}$
- True positive rate: $TP / \text{actual yes}$
- False positive rate: $FP / \text{actual no}$
- True negative rate: $TN / \text{actual no}$

The validation summary statistics are provided in the table below. They show that a threshold of 2 or 5 mmWE / day provides the highest accuracy of the thresholds tested for both RACMO and MAR, with lesser accuracy for lower and higher thresholds. Since the overall prevalence of melt is relatively low (10.21%) across all grid cells, a high true positive rate is important to demonstrate that the models can capture melt when it actually does occur, while maintaining a low false positive rate. Because the true positive rate for the 2 mmWE / day threshold is ~12–15% higher than the 5 mmWE / day threshold, with a false positive rate only ~1–2% higher, we consider 2 mmWE / day the optimal threshold value for classification of a “melt day” in the models.

Melt threshold (mmWE / day)	Metric	RACMO	MAR
0.1	accuracy	92.24%	93.26%
1	accuracy	93.31%	94.21%
2	accuracy	93.86%	94.50%
5	accuracy	94.18%	94.43%
8.25	accuracy	93.17%	93.71%
10	accuracy	92.50%	93.24%
0.1	true positive rate	82.58%	82.17%
1	true positive rate	75.84%	74.61%
2	true positive rate	71.02%	69.74%
5	true positive rate	56.36%	57.18%
8.25	true positive rate	38.49%	44.24%
10	true positive rate	30.14%	37.77%
0.1	false positive rate	6.66%	5.47%
1	false positive rate	4.70%	3.57%
2	false positive rate	3.54%	2.69%
5	false positive rate	1.52%	1.34%
8.25	false positive rate	0.62%	0.67%
10	false positive rate	0.41%	0.46%
0.1	true negative rate	93.34%	94.53%
1	true negative rate	95.30%	96.43%
2	true negative rate	96.46%	97.31%
5	true negative rate	98.48%	98.66%
8.25	true negative rate	99.38%	99.33%
10	true negative rate	99.59%	99.54%

Summary validation statistics for the range of “melt day” thresholds applied to RACMO and MAR data.

Finally, to evaluate how well RACMO and MAR simulate the spatial pattern of NE Greenland melt, we created maps of the overall percentage of JJA days with melt for each dataset and their differences with the MEaSUREs data. The plots for the 1, 2, and 5 mmWE / day thresholds are shown below. These maps show that RACMO and MAR broadly agree with MEaSUREs on the spatial pattern of mean JJA melt. RACMO overestimates the frequency of melt in most areas with a 1 mmWE / day threshold and underestimates melt prevalence with a 5 mmWE / day threshold, with a relatively even spatial distribution of overestimation and underestimation for the 2 mmWE / day threshold. MAR has a more definite spatial pattern to its bias, with an underestimation of melt frequency at higher elevations even for the 1 mmWE / day threshold, and an overestimation of melt frequency at lower elevations even for the 5 mmWE / day threshold. These results show that the models generally simulate the melt area in NE Greenland accurately when an appropriate threshold is found for comparison with independent passive

microwave satellite data, and that RACMO is more consistent than MAR in its agreement with the satellite melt data across elevation ranges.

Top row: mean percentage of JJA days with melt from RACMO and MAR for a “melt day” threshold of 1 mmWE / day, with MEaSUREs passive microwave melt frequency included for comparison. Bottom row: differences in JJA melt day frequency between the datasets.

As in previous figure, but for a “melt day” threshold of 2 mmWE / day.

As in previous two figures, but for a “melt day” threshold of 2 mmWE / day.

My recommendation is to provide the authors with the opportunity to make a much stronger case for the combined importance of atmospheric rivers and foehn events.

Specific Comments:

1. Does one get different results for the ARs and their extremes if ERA5 rather than MERRA2 is used for the integrated vapor transport?

This is a good question. AR detection frequency based on IVT is affected by the choice of reanalysis although the degree of difference changes for each detection algorithm (Collow et al., 2022). We tested a version of the Mattingly et al., 2020 AR detection forced with ERA5 IVT to see if there are differences in detection frequency and impacts. The AR frequency from 1980–2020 for JJA is very similar between the ERA5 and MERRA2 versions of the algorithm (the detection frequency in the ERA5 version is ~2 percentage points higher). We added a sentence to the Methods to indicate that we found negligible difference in AR detection frequency and AR-related melting. A version of Figure 2 with AR detection from the ERA5 version is shown below.

Percentage of surface melt attributable to atmospheric rivers. (a–c) The percentage of RACMO2 summer (JJA) daily surface melt and (d–f) the percentage of surface melt at or above the 99th percentile of the 1980–2020 monthly mean climatology that occurred (a), (d) the same day, (b), (e) 24 hours later, and (c), (f) 48 hours later of an AR landfall in NW Greenland (orange outline in Fig. 1a). AR detection done using ERA5 version of algorithm from Mattingly et al., 2020.

2. Figure 5 needs NEGIS like Figure ED5.

The NEGIS outline has been added to Figure 5a.

3. Figure ED2: What is the horizontal scale of the MODIS imagery? I am not sure what is bare ice and melt ponds in the pictures.

We've added a horizontal scale to Figure ED3 (Figure ED2 in previous version) along with labels to help identify the bare ice and melt pond features.

4. Figure ED3 (and ED2): Aren't the shortwave anomalies due to foehn clearance on the lee side important?

It's true that the cloud clearance effect from the foehn is an important aspect of melt that is observed on the Antarctic Peninsula and not mentioned in the manuscript. We've added an analysis of the downward shortwave radiation anomalies to Figures ED3 and ED4 (Figures ED2 and ED3 in the previous version). They show that downward shortwave anomalies are generally neutral to slightly positive in NE Greenland during AR events. This indicates that the AR-related clouds and moisture remain concentrated on the western side of the ice sheet and the foehn clearance effect on the lee side is occurring which contributes to the surface melting. We've added text to the "Intense northeast Greenland summer melt following western Greenland atmospheric rivers" section to mention the cloud clearance effect along with citations connecting it with similar observations from the Antarctic Peninsula.

5. Figure ED4: It is surprising that the relative humidity didn't drop more sharply. Often foehns are very dry on the lee side.

This is an interesting observation. Although the RH did decrease quickly by ~20% at both stations on 19 July as the foehn developed, RH values were not extremely low throughout the remainder of the event. Our foehn detection criteria do not include any absolute threshold for RH, rather they identify RH values that decrease quickly and are relatively low compared to the local climatology. It is possible that drying in NE Greenland during foehn events is not as intense as in other regions where foehn has been studied, due to the relatively gentle slope of the GrIS in comparison to more steep and narrow mountain ranges in places such as the Antarctic Peninsula or the Alps. This would be an interesting area for further study.

6. Figure ED10: I didn't get much out of this figure to support the case being made. Why use melt from MAR here?

This figure was a remnant of a previous iteration of the work that used self-organizing map analysis more extensively to classify the direction of moisture transport toward NE Greenland. However, we agree with the reviewer that this figure is superfluous and the disproportionate influence of AR90 events on melt can be determined from the existing figures. Therefore we have removed the original Figure ED10, and also removed the description of the self-organizing map method from the Methods section.

Reviewer #2 (Remarks to the Author):

General: This is a well-written paper and quite complete. I do have a few questions:

1) Figure 3 shows increasing melt with higher IVT but also suggests that there is still some melt even at low IVT. Presumably there is some melt at the edges of the ice sheet at the height of summer simply because it gets warm. It would be interesting to look at the relationship in the wings of the summer season.

Thank you for the suggestion. It is true that simply the presence of solar radiation with slightly enhanced moisture transport is enough to trigger low magnitude melt during the summer. Melting triggered by foehn during winter months has been observed over the Antarctic Peninsula (Kuipers Munneke et al., 2018; Wille et al., 2019), so it is reasonable to expect a similar dynamic exists over the NE Greenland. We explored the IVT magnitude / total melt relationship for other months besides JJA and found that melt in this region becomes negligible in every other month. For instance, the plot below for May demonstrates very minimal melt values. We've added a sentence to the "Intense northeast Greenland summer melt following western Greenland atmospheric rivers" section to describe this.

Relationship between May AR intensity and magnitude of surface melt. Box and whisker plot shows the maximum integrated water vapor transport (IVT) of NW Greenland landfalling ARs (orange box in Fig. 1a) detected using the algorithm in (a) Mattingly et al., 2020 and (b) Wille et al., 2021 and the corresponding daily surface melt 24 hours after the initial AR landfall summed over the NE Greenland domain (red box in Fig. 1a). Data sources are MERRA-2 for IVT and RACMO2 for surface melt.

Even when breaking down the summer months on a per month basis, we see that the most significant melt occurs in June and July with a large decrease in August as seasonal temperatures begin to drop. Overall, this underscores how solar radiation is a necessary ingredient on top of large sensible heat fluxes for melting in this region.

Relationship between June, July, and August AR intensity and magnitude of surface melt. Box and whisker plot shows the maximum integrated water vapor transport (IVT) of NW Greenland landfalling ARs (orange box in Fig. 1a) detected using the algorithm in (a) Mattingly et al., 2020 and (b) Wille et al., 2021 and the corresponding daily surface melt 24 hours after the initial AR landfall summed over the NE Greenland domain (red box in Fig. 1a). Data sources are MERRA-2 for IVT and RACMO2 for surface melt.

2) The case in Figure 1 shows an AR originating over Canada. This seems a little rare when looking at Hermann et al. (ref below) where they show back trajectories mostly from the south and not that many linked to melting in the NE.

This is an interesting point. We have not performed any systematic analysis of the origin or direction of approach of ARs to NW Greenland, as these aspects are outside the scope of our study. The July 2014 event in Figure 1 is an example chosen more or less randomly from the post-2012 period in which both the KPC_L and KPC_U stations have data available, and may not represent a typical AR origin location. However, it does appear that the track of this example event – with an AR originating near Hudson Bay and tracking to the northeast over Baffin Bay toward northwest Greenland – is consistent with at least a subset of the trajectories in Hermann et al. 2020. Their Figure 2 appears to show a substantial number of trajectories passing across Hudson Bay and progressing toward Baffin Bay, although this air mass pathway is not labeled as a representative cluster in their paper. Rather, this pathway of air masses toward northern Greenland is located in between the “N1” and “N2” representative trajectories.

Additionally, Hermann et al. 2020 analyzed ice-sheet-wide large melt events, while our study is focused on NE Greenland melt. There may be some differences in the characteristics of air mass origin and transport for these more regional events, and NE Greenland may sometimes experience enhanced melt that is not coincident with extensive melt throughout all of Greenland (this is the case for the example event in Figure 1). This may help explain why the air mass pathway for this event does not appear as a trajectory cluster for large-scale Greenland melt events in Hermann et al. 2020.

3) Are there ever Foehn conditions to the east without an AR in the NW?

Figure 6 shows that NE Greenland foehn conditions occur without an AR90 in the NW 14.97% of the time in JJA. The equivalent percentage when considering *all* ARs is 9.89%. This indicates that foehn does occur without ARs in the NW, which is to be expected given that AR events are relatively infrequent – around 34% of JJA 6-hourly timesteps have an AR of any intensity in the NW Greenland domain during the preceding 48 hours. However, the most intense foehn-related melt rates occur most frequently during AR90 events.

Specific:

Line 113: Reference is to Basin 8 – This is the only reference to a basin number?

This was left over from a previous iteration of the paper that reported results with reference to the 8 large-scale GrIS glacial drainage basins. It has been removed. Thank you for pointing this out.

Lines 124-127: Sometimes, in cases of a mountain range with a flat plane to the east, lee waves associated with the foehn effect can lower the surface pressure at the base of the mountain range drawing cooler air from the east towards the base of the mountains. A higher resolution model might show this.

This is an interesting point. We see some evidence of lee-wave generation at higher altitudes from plotting the RH and vertical wind speed at around 2000m ASL, but are unsure that this is the generation of the cold air, rather it may be the inability of the foehn to erode the cold pool which already existed. At the base of the mountains, we do not detect any evidence of inflow of cold air from the east, but it is possible that higher vertical resolution is required to see this. We have added text to the paper noting the possibility of lee wave influence and that it could be investigated with higher resolution simulations.

Line 381: This link does not work:

<https://www.earthsystemgrid.org/dataset/ucar.cgd.cesm4.artmip.tier1.html>.

The link to the dataset was updated during the review process. We replaced the old link with the correct link below.

https://www.earthsystemgrid.org/dataset/ucar.cgd.artmip.tier2.catalogues.merra2.native.wille_v1_VT.html

Figures:

Figure 2: The color scheme here leaves something to be desired. Perhaps putting white contour lines at, say 50% and 75%, would make it easier to see differences in the results.

Good point. We tried adding white contours at 50% and 75%, but it makes the figure very messy to read. The main issue with this figure was that we were not utilizing the full color scale since there were hardly any values below 40%. So we edited the range of plotted values so that everything below 40% is given a yellow color and the full color scale is now used for values above 40%. The new figure 2 is in the manuscript. It is now more clear to see where the AR-related foehn winds have the greatest impact.

Reference:

Hermann, M., L. Papritz, and H. Wernli (2020), A Lagrangian analysis of the dynamical and thermodynamic drivers of large-scale Greenland melt events during 1979–2017, *Weather Clim. Dynam.*, 1(2), 497-518, doi:10.5194/wcd-1-497-2020.

References

- Alexander, P. M., Tedesco, M., Koenig, L. & Fettweis, X. Evaluating a Regional Climate Model Simulation of Greenland Ice Sheet Snow and Firn Density for Improved Surface Mass Balance Estimates. *Geophys. Res. Lett.* 46, 12073–12082 (2019). <https://doi.org/10.1029/2019GL084101>
- Collow, A. B. M., Shields, C. A., Guan, B., Kim, S., Lora, J. M., McClenny, E. E., et al. (2022). An Overview of ARTMIP's Tier 2 Reanalysis Intercomparison: Uncertainty in the Detection of Atmospheric Rivers and Their Associated Precipitation. *Journal of Geophysical Research: Atmospheres*, 127(8), e2021JD036155. <https://doi.org/10.1029/2021JD036155>
- Fettweis, X., Tedesco, M., van den Broeke, M., & Ettema, J. (2011). Melting trends over the Greenland ice sheet (1958–2009) from spaceborne microwave data and regional climate models. *The Cryosphere*, 5(2), 359–375. <https://doi.org/10.5194/tc-5-359-2011>
- Kimball, J. S., Du, J., Meierbachtol, T. W., Kim, Y., & Johnson, J. V. (2021). Comparing Greenland Ice Sheet Melt Variability From Different Satellite Passive Microwave Remote Sensing Products Over a Common 5-year Record. *Frontiers in Earth Science*, 9, 654220. <https://doi.org/10.3389/feart.2021.654220>
- Kuipers Munneke, P., Luckman, A. J., Bevan, S. L., Smeets, C. J. P. P., Gilbert, E., van den Broeke, M. R., et al. (2018). Intense Winter Surface Melt on an Antarctic Ice Shelf. *Geophysical Research Letters*, 45(15), 7615–7623. <https://doi.org/10.1029/2018GL077899>
- Mioduszewski, J. R., Rennermalm, A. K., Hammann, A., Tedesco, M., Noble, E. U., Stroeve, J. C., & Mote, T. L. (2016). Atmospheric drivers of Greenland surface melt revealed by self-organizing maps. *Journal of Geophysical Research: Atmospheres*, 121(10), 5095–5114. <https://doi.org/10.1002/2015JD024550>
- Mote, T. L. (2014). MEASUREs Greenland Surface Melt Daily 25km EASE-Grid 2.0, Version 1. Boulder, Colorado USA: National Snow and Ice Data Center Distributed Active Archive Center. Retrieved from <https://doi.org/10.5067/MEASURES/CRYOSPHERE/nsidc-0533.001>. Accessed 5 Dec 2022.
- Wille, J. D., Favier, V., Dufour, A., Gorodetskaya, I. V., Turner, J., Agosta, C., & Codron, F. (2019). West Antarctic surface melt triggered by atmospheric rivers. *Nature Geoscience*, 12(11), 911–916. <https://doi.org/10.1038/s41561-019-0460-1>

REVIEWERS' COMMENTS

Reviewer #1 (Remarks to the Author):

Thanks to the authors for their careful and comprehensive responses to the issues raised in my review, especially the comparisons with satellite passive microwave melt areas. RACMO2 is the model

to apply, clearly. Their arguments for the importance of ARs and foehn are now much more convincing, as well as their increasing impact in the future. My recommendation is acceptance.

Reviewer #2 (Remarks to the Author):

The authors have addressed my concerns and it should be published.

I expect this paper will stimulate even more research into the large scale forcing of melt events over Greenland.

Reviewer #1 (Remarks to the Author):

Thanks to the authors for their careful and comprehensive responses to the issues raised in my review, especially the comparisons with satellite passive microwave melt areas. RACMO2 is the model to apply, clearly. Their arguments for the importance of ARs and foehn are now much more convincing, as well as their increasing impact in the future. My recommendation is acceptance.

Reviewer #2 (Remarks to the Author):

The authors have addressed my concerns and it should be published. I expect this paper will stimulate even more research into the large scale forcing of melt events over Greenland.

Thanks to both reviewers for taking the time to engage carefully with our work. Their helpful comments and feedback improved the quality of the final paper.